# Mathematical Modeling of Andrographolide Therapy Effects and Immune Response in In Vivo Dynamics of SARS-CoV-2 Infection

**DOI:** 10.3390/v17070891

**Published:** 2025-06-25

**Authors:** Panittavee Yarnvitayalert, Teerapol Saleewong

**Affiliations:** Department of Mathematics, King Mongkut’s University of Technology Thonburi, Bangkok 10140, Thailand; panittavee.n@mail.kmutt.ac.th

**Keywords:** andrographolide, COVID-19, mathematical model, pharmacokinetics, pharmacodynamics, viral dynamic model

## Abstract

This study explores the viral dynamics of SARS-CoV-2 infection within host cells by incorporating the pharmacological effects of andrographolide—a bioactive compound extracted from *Andrographis paniculata*, renowned for its antiviral, anti-inflammatory, and immunomodulatory properties. Through the application of mathematical modeling, the interactions among the virus, host cells, and immune responses are simulated to provide a comprehensive analysis of viral behavior over time. Two distinct models were employed to assess the impact of varying andrographolide dosages on viral load, target cell populations, and immune responses. One model revealed a clear dose–response relationship, whereas the other indicated that additional biological or pharmacological factors may modulate drug efficacy. Both models demonstrated stability, with basic reproductive numbers (R_0_) suggesting the potential for viral propagation in the absence of effective therapeutic interventions. This study emphasizes the significance of understanding the pharmacokinetics (PK) and pharmacodynamics (PD) of andrographolide to optimize its therapeutic potential. The findings also underscore the necessity for further investigation into the compound’s absorption, distribution, metabolism, and excretion (ADME) characteristics, as well as its prospective applications in the treatment of not only COVID-19 but also other viral infections. Overall, the results lay a foundational framework for future experimental research and clinical trials aimed at refining andrographolide dosing regimens and improving patient outcomes.

## 1. Introduction

The COVID-19 pandemic has precipitated a global health crisis, intensifying the need for effective therapeutic interventions against SARS-CoV-2. While conventional antiviral drugs have played a crucial role in the management of viral infections, their limitations—such as resistance, limited efficacy, and adverse effects—have prompted increased interest in alternative therapies, including traditional herbal medicines.

One notable example is *Andrographis paniculata* (Burm. f.) Wall. ex Nees, a well-established medicinal plant widely used in several Asian countries, including Thailand, China, and India. In 2021, Thailand’s Department of Medical Services approved the use of *Andrographis paniculata* extract (APE) for the early treatment of asymptomatic COVID-19 patients without high-risk comorbidities [1]. The primary bioactive compound in this plant, andrographolide, has demonstrated the ability to inhibit the intracellular replication of SARS-CoV-2 [2].

APE has been reported to exert antiviral effects by interfering with viral membrane fusion processes, potentially through its interactions with the SARS-CoV-2 spike protein–angiotensin-converting enzyme 2 (ACE2) complex, as well as direct interactions between the spike protein and host cell ACE2 receptors [2]. These findings suggest a promising role for andrographolide in the therapeutic landscape of COVID-19 and warrant further investigation through integrative approaches, such as mathematical modeling.

During the COVID-19 pandemic, *Andrographis paniculata* emerged as a candidate antiviral treatment with multiple proposed mechanisms of action, including the inhibition of viral replication, the modulation of immune responses, and the attenuation of inflammatory processes. A pilot study conducted in Thailand [3], involving six patients with mild COVID-19 symptoms, reported promising outcomes. The patients received andrographolide extract at a dosage of 180 mg/day in conjunction with standard supportive care over five consecutive days. The treatment led to significant reductions in the severity of COVID-19-related symptoms—particularly cough and headache—on days 3 and 5, with three patients testing negative by RT-PCR on day 5 [3]. While previous research supports the therapeutic potential of a 180 mg/day dosage of andrographolide in treating respiratory infections [4], its specific effects on the in vivo dynamics of SARS-CoV-2 infection remain insufficiently understood.

This study aims to evaluate the efficacy of andrographolide in mitigating SARS-CoV-2 infection through the application of mathematical modeling. By integrating principles of pharmacokinetics and pharmacodynamics (PK/PD) with viral dynamics, the model examines the influence of andrographolide on disease progression, viral load, and host immune responses. The key variables considered include target cell availability, viral replication rate, immune activation, and the duration of viral shedding.

Beyond model construction, the study seeks to validate its findings against clinical data. Drawing on previous modeling approaches—such as that of Gonçalves et al., who identified critical parameters for predicting antiviral treatment outcomes in SARS-CoV-2 infection [5], and Dodds et al., who explored pharmacological interventions targeting different stages of the viral life cycle [6]—this research builds a comprehensive framework for assessing therapeutic strategies.

Ultimately, by elucidating the dynamic interactions among andrographolide, the virus, and the host immune system, this work aims to contribute to a deeper understanding of *Andrographis paniculata*—a potential therapeutic intervention in the ongoing effort to combat COVID-19.

## 2. Mathematical Model

The mathematical model in this study integrates pharmacokinetics/pharmacodynamics (PK/PD) with viral dynamics to explore the impact of andrographolide on SARS-CoV-2 infection. The PK/PD model tracks changes in andrographolide concentration, while the viral dynamic model examines alterations in target cells, virus, and immune response, incorporating andrographolide’s effects. This combined approach provides insights into andrographolide’s mechanism of action and its potential for treating COVID-19 [7].

### 2.1. Andrographolide’s Pharmacokinetic/Pharmacodynamic Model

The PK/PD model provides insights into the changes in andrographolide concentration over time, allowing us to discern how the drug is metabolized and how its levels correlate with therapeutic efficacy [8]. This model elucidates the pharmacokinetic processes governing the absorption, distribution, metabolism, and excretion of andrographolide, as well as its pharmacodynamic effects on the body. The mathematical equation describing the PK model of andrographolide is as follows:(1)$$\begin{array}{c} \frac{dC_{p}}{dt}=dose-k_{10}C_{p}-k_{12}C_{p}+k_{21}C_{t},\\ \frac{dC_{t}}{dt}=k_{12}C_{p}-k_{21}C_{t} \end{array}$$
where the variable and the parameter are shown in Table 1.

The pharmacokinetic (PK) profiles of the medication were simulated based on established parameter distributions reported in the literature. This simulation elucidates the dynamics of the drug’s absorption, distribution, metabolism, and excretion over time. The pharmacodynamic (PD) effectiveness of the medication, which delineates the relationship between the concentration of the drug in whole blood and its corresponding therapeutic effects (as shown in Figure 1), was quantified through the application of Equation (1).

Efficacy, defined as the drug’s capacity to elicit the desired therapeutic response at the appropriate dosage, is a critical consideration in evaluating pharmacological agents. For andrographolide, a bioactive compound recognized for its potential health benefits, its efficacy was systematically assessed using a system of mathematical equations that incorporated variables such as blood concentration, duration of exposure, and specific clinical outcomes. These analytical models enabled researchers to ascertain the performance of andrographolide in achieving its intended effects in patients, thereby informing optimal dosing regimens and treatment strategies. Consequently, the efficacy of andrographolide, denoted as ϵ, was evaluated based on the plasma concentration of andrographolide ($C_{p}$) and its half-maximal inhibitory concentration (${IC}_{50}$) [2,5]. The relationship defining the efficacy of andrographolide is presented in the following equations:(2)$$\epsilon (t)=\frac{C_{p}(t)}{C_{p}(t)+{IC}_{50}}$$

In Thailand, the Ministry of Public Health recommends a 5-day course of andrographolide at 180 mg t.i.d. (three times daily) [9]. Therefore, the average efficacy of andrographolide ($\bar{\epsilon }$) during this 5-day treatment period is represented by the following equation:(3)$$\bar{\epsilon }(t)=\frac{1}{5}\int_{0}^{5}\frac{C_{p}(t)}{C_{p}(t)+{IC}_{50}}dt.$$
where $0\leq \bar{\epsilon }\leq 1$.

Given that andrographolide suppressed the production of infectious SARS-CoV-2 virions, its inhibitory impact was integrated into the proposed viral dynamic model.

### 2.2. Modeling the Dynamics of SARS-CoV-2 Infection

To model the virus life cycle and immune response, we employed two viral dynamic models that represent different levels of immune complexity. Model 1 describes early viral dynamics, incorporating a nonspecific innate immune response, while Model 2 builds upon this by including specific lytic and non-lytic immune responses, reflecting a more detailed adaptive immune mechanism. These two models allow us to examine how different immune pathways and antiviral treatments affect viral load over time.

In Model 1, uninfected target cells (T) become infected ($I_{1}$) by the virus (V) at a rate of βTV, while the infection process is inhibited by the immune response at a rate of σTF. Following infection, the target cells survive the incubation period and transition to productively infected cells ($I_{2}$) at a rate of k, where 1/k represents the average transition time from $I_{1}$ to $I_{2}$.

New virus production (V) occurs through RNA synthesis from productively infected cells ($I_{2}$) at a rate of p. The process of RNA replication, which leads to the formation of new viruses, can be inhibited by the mean effectiveness of andrographolide ($\bar{\epsilon }$), as determined from the PK/PD model. To account for this inhibitory effect, the viral production rate p is multiplied by ($1-\bar{\epsilon }$). Productively infected cells undergo cell death at a rate of δ, where 1/δ represents the average lifespan of these cells.

Newly generated viruses can interact with and infect additional target cells, while viral clearance takes place at a rate of c. The immune cell population (F) proliferates at a rate of g and undergoes natural decay at a rate of ω. The structure of Model 1 is depicted in Figure 2. The system of equations representing Model 1 is presented as follows:(4)$$\begin{array}{l} \frac{dT}{dt}=-\beta TV-\sigma TF,\\ \frac{dI_{1}}{dt}=\beta TV-kI_{1},\\ \frac{dI_{2}}{dt}=kI_{1}-\delta_{I_{2}}I_{2}\\ \frac{dV}{dt}=(1-\bar{\epsilon })pI_{2}-cV,\\ \frac{dF}{dt}=gI_{1}-\omega F. \end{array}$$
where the variables and parameters are shown in Table 2.

While Model 1 captures the dynamics of viral infection and the nonspecific innate immune response, it does not explicitly represent specific immune mechanisms. Therefore, Model 2 extends this framework by incorporating both lytic and non-lytic immune responses, providing a more comprehensive understanding of host–virus interactions.

In Model 2, we consider the specific lytic and non-lytic immune responses by using the compartment model in Figure 3. This model incorporates viral replication, which is inhibited by the immune response at a rate of σF+1. It is assumed that the target cells proliferate at a rate of λ and undergo natural cell death at a rate of $\delta_{T}$ [10]. The mathematical model representing the life cycle of SARS-CoV-2 is formulated as follows:(5)$$\begin{array}{c} \frac{dT}{dt}=\lambda -\delta_{T}-\frac{\beta \;TV}{\sigma F+1},\\ \frac{dI_{1}}{dt}=\frac{\beta \;TV}{\sigma F+1}-kI_{1},\\ \frac{dI_{2}}{dt}=kI_{1}-\delta_{I_{2}}I_{2}\\ \frac{dV}{dt}=(1-\bar{\epsilon })pI_{2}-cV,\\ \frac{dF}{dt}=gI_{1}-\omega F. \end{array}$$
where the variables and parameters are shown in Table 2.

The viral dynamic models, Model 1 and Model 2, describe SARS-CoV-2 infection within a host by quantifying target cells, virion cells, and immune response over time t. These models achieve this by fitting their numerical solutions to the dataset.

### 2.3. Dataset

This study is exploratory in nature and utilized viral load data from four patients infected with SARS-CoV-2 who were under observation in a hospital in Thailand [4] to perform parameter estimation. These patients had mild symptoms and received 180 mg of andrographolide three times daily for five days. Viral loads in nasopharyngeal swabs were measured at different time points. Viral loads in nasopharyngeal swabs were measured at different time points, specifically on days 3, 5, and 7 post-infection [4].

## 3. Numerical Results of Viral Dynamic Model

In the fitting process, the numerical results of the viral dynamic models (Model 1 and Model 2) in Equations (4) and (5) were fitted to a dataset [4] to estimate the parameters and the results of the parameters of the four patients, as shown in Table 3 and Table 4. In these results, the behavior of the SARS-CoV-2 dynamic model focuses on the quantity of viral load, duration of viral shedding, quantity of epithelial cells infected, and immune response, as reflected in the graph of target cells, viral load, and immune response in Figure 4.

The viral dynamics in Models 1 and 2 exhibit distinct patterns in the progression and interaction of target cells, viral load, and immune response during SARS-CoV-2 infection.

In Model 1, there is a rapid decline in the target cell population within the first 5 days, followed by stabilization by day 7. This initial decline likely reflects the virus’s rapid infection and destruction of susceptible cells. The viral load increases sharply and peaks between days 2 and 5, indicating a period of high viral replication and spread, which corresponds with the typical peak of symptoms in viral infections. After this peak, the viral load decreases and is completely cleared by day 15, suggesting effective viral clearance mechanisms. The immune response is activated immediately following infection, peaking shortly after the viral load reaches its maximum, and then gradually declines, resolving completely by day 20. This pattern indicates a robust immune response capable of clearing the virus and leading to full recovery.

Conversely, Model 2 shows a more gradual decline in the target cell population, with minimal infection observed. The viral load drops rapidly at the onset and stabilizes within one day, suggesting effective containment or a less aggressive infection. The immune response peaks on the first day, potentially indicating a rapid immune activation, possibly due to pre-existing immunity or a less virulent viral strain. After peaking, the immune response decreases and stabilizes by day 12. The faster stabilization and lower overall immune activity in Model 2 compared to Model 1 may suggest a scenario where the infection is controlled more rapidly with less immune system involvement. This could result from factors such as a lower viral load, reduced viral pathogenicity, or a more efficient innate immune response.

The differences between the two models highlight various possible infection scenarios and suggest that andrographolide dosage may significantly impact treatment outcomes. Studies have shown that andrographolide has therapeutic benefits for respiratory infections, including influenza, URTIs, pharyngo-tonsillitis, and more recently, COVID-19. The efficacy of andrographolide appears to be dose dependent, with higher doses generally leading to more pronounced symptom relief and greater patient satisfaction.

For instance, Kulichenko demonstrated that low doses of andrographolide (30 mg and 45 mg per day) could effectively reduce the duration of influenza [12]. Saxena found that a slightly higher dose (60 mg per day) was effective in alleviating symptoms of uncomplicated URTIs [13]. Thamlikitkul observed that higher doses (180 mg and 360 mg per day) not only alleviated symptoms like fever and sore throat but also led to greater patient satisfaction [14]. Kulthanit proposed using 180 mg per day for COVID-19 treatment, achieving complete clinical recovery within 5 days, suggesting significant potential for andrographolide in managing SARS-CoV-2 infections [9]. However, clinical use faces limitations, including variability in patient responses, potential side effects, and the need for more comprehensive trials to establish standardized dosing protocols and safety profiles.

To determine the viral load related to andrographolide dosage—specifically, 30, 45, 60, 180, and 360 mg per day—we incorporated pharmacokinetic (PK) profiles to estimate the effective drug exposure (ϵ), which was subsequently used in the model simulations to assess dose-dependent antiviral effects [11]. The mean efficacy of andrographolide, $\bar{\epsilon }$, is presented in Table 5 [11]. The numerical simulations of viral dynamics with andrographolide therapy and immune responses (Model 1 with nonspecific innate immune response and Model 2 with lytic and non-lytic immune responses) provide insights into the effects of andrographolide dosage, as shown in Figure 5. In Model 1, target cell numbers declined and stabilized around day 10, while viral load and immune response peaked around day 5. The peak levels of viral load and immune response varied with andrographolide dosages, indicating that the drug’s effectiveness is dose dependent. Higher doses generally led to greater reductions in viral load and stronger immune responses, underscoring the importance of adequate dosing for optimal therapeutic effects against SARS-CoV-2.

In contrast, Model 2 showed that target cell count and viral load decreased rapidly initially and stabilized by day 5. The immune response peaked on the first day, then decreased and stabilized by day 15. Interestingly, different andrographolide dosages did not significantly alter the trends observed in Model 2. This suggests that the immune response and viral dynamics in Model 2 are less sensitive to variations in andrographolide dosage, or that the drug’s effects are less pronounced in this scenario. This could be due to differences in the underlying assumptions and mechanisms in Model 2 compared to Model 1.

These findings emphasize the importance of considering both viral dynamics and andrographolide dosage when developing treatment strategies for SARS-CoV-2 and other respiratory infections. The simulations demonstrate that different andrographolide dosages directly influence the viral kinetics and immune response within each model, affecting not only the peak viral load but also the timing and magnitude of immune activation. The observed differences in model responses highlight the need for personalized treatment approaches and further research to optimize andrographolide’s therapeutic use. Given that drug effects substantially alter system behavior, a stability analysis was subsequently performed to evaluate the conditions under which viral clearance can be achieved. This allows us to identify threshold parameters that ensure infection control, reinforcing the translational relevance of model-based treatment predictions.

## 4. Stability of Viral Dynamic Model

Viral dynamic models are used to describe the interaction between a virus and the host’s immune response, often focusing on the infection and replication processes within an individual host. These models are crucial for understanding the progression of viral infections and the impact of antiviral treatments. Stability analysis in viral dynamic models involves examining the behavior of the system.

### 4.1. Stability of Model 1

To analyze stability, one first finds the equilibrium points by setting the derivatives to zero, as follows:(6)$$\frac{dT}{dt}=\frac{dI_{1}}{dt}=\frac{dI_{2}}{dt}=\frac{dV}{dt}=\frac{dF}{dt}=0.$$

The equilibrium point $\left(T^{*},\;I_{1}^{*},\;I_{2}^{*},\;V^{*},\;F^{*}\right)$ is as follows:(7)(0,0,0,0,0).

The Jacobian matrix J is given by the following:(8)$$J=\left[\begin{array}{ccccc} -\beta V-\sigma F & 0 & 0 & -\beta T & -\sigma T\\ \beta V & -k & 0 & \beta T & 0\\ 0 & k & -\delta_{I_{2}} & 0 & 0\\ 0 & 0 & (1-\bar{\epsilon })p & -c & 0\\ 0 & g & 0 & 0 & -\omega \end{array}\right]$$

By substituting the parameters from Table 3 to evaluate the eigenvalues using det(J−λI)=0, all eigenvalues are less than or equal to zero. Therefore, the equilibrium point is stable [15].

### 4.2. Stability of Model 2

Consider equilibrium point of Model 2 by setting the derivative to be equal to zero, as follows:(9)$$\frac{dT}{dt}=\frac{dI_{1}}{dt}=\frac{dI_{2}}{dt}=\frac{dV}{dt}=\frac{dF}{dt}=0.$$

The equilibrium point $\left(T^{*},\;I_{1}^{*},\;I_{2}^{*},\;V^{*},\;F^{*}\right)$ is as follows:(10)$$\left(\frac{c\delta_{I_{2}}\sigma g(\lambda -\delta_{T})-\omega k}{\omega k\beta p(1-\bar{\epsilon })},\frac{\lambda -\delta_{T}}{k},\frac{\lambda -\delta_{T}}{\delta_{I_{2}}},\frac{p(1-\bar{\epsilon })(\lambda -\delta_{T})}{\delta_{I_{2}}c},\frac{g(\lambda -\delta_{T})}{\omega k}\right).$$

The Jacobian matrix J is given by the following:(11)$$J=\left[\begin{array}{ccccc} -\frac{\beta V}{\sigma F+1} & 0 & 0 & -\frac{\beta T}{\sigma F+1} & \frac{\sigma \beta VT}{{\left(\sigma F+1\right)}^{2}}\\ \frac{\beta V}{\sigma F+1} & -k & 0 & \frac{\beta T}{\sigma F+1} & -\frac{\sigma \beta VT}{{\left(\sigma F+1\right)}^{2}}\\ 0 & k & -\delta_{I_{2}} & 0 & 0\\ 0 & 0 & (1-\bar{\epsilon })p & -c & 0\\ 0 & g & 0 & 0 & -\omega \end{array}\right]$$

By substituting the parameters from Table 4 to evaluate the eigenvalues through det(J−λI)=0, all eigenvalues have a negative real part. Thus, the equilibrium point is stable [15].

A stability analysis of the viral dynamic models indicates that the behavior of these models is stable under the specified parameter values. This suggests that, in the absence of external perturbations or significant changes in key parameters, the system will tend to return to equilibrium, reflecting predictable and biologically plausible infection dynamics. Additionally, understanding the potential spread and severity of an infection within a host is informed by the within-host basic reproductive number, $R_{0}$, which provides a threshold criterion for determining whether the virus will persist or be cleared from the system.

## 5. The Basic Reproductive Number

The within-host basic reproductive number, $R_{0}$, represents the average number of newly infected target cells generated by a single infected cell within the host. In these models, $R_{0}$ explains the potential of the SARS-CoV-2 virus to either establish an infection at the cellular level within the host.

### 5.1. The Basic Reproductive Number of Model 1

Consider the viral dynamic model with andrographolide and immune response (Model 1). $R_{0}$ of Model 1 is computed by adopting the next-generation matrix (NGM) technique [16]. From Equation (4), we are concerned with the populations that spread the infection. We need to represent the exposed, V, and infected, $I_{1}$ and $I_{2}$, classes. The kinetic model is defined by using the following equations:(12)$$\begin{array}{c} \;\frac{dI_{1}}{dt}=\beta TV-kI_{1},\\ \frac{dI_{2}}{dt}=kI_{1}-\delta_{I_{2}}I_{2}\\ \frac{dV}{dt}=(1-\bar{\epsilon })pI_{2}-cV, \end{array}$$

For this system, we consider $\frac{dX}{dt}=F_{i}(x)-V_{i}(x)$, where $X={\left[I_{1},I_{2},V\right]}^{t}$, $F_{i}(x)\;$ is the matrix of rate of appearance of new infections in a compartment, $V_{i}(x)\;$ is the matrix of rate of other transitions between one compartment and other infected compartments. Then we consider $F=\left(\frac{\partial F_{i}(x_{0})\;}{\partial x_{j}\;}\right)\;$ and $V=\left(\frac{\partial V_{i}(x_{0})\;}{\partial x_{j}\;}\right).\;$ The next generation matrix is given by the following:(13)$${FV}^{-1}=\left(\begin{array}{ccc} 0 & 0 & \frac{\beta T_{0}}{c}\\ 0 & 0 & 0\\ \frac{(1-\bar{\epsilon })p}{\delta_{I_{2}}} & \frac{(1-\bar{\epsilon })p}{\delta_{I_{2}}} & 0 \end{array}\right).$$

Consider $det({FV}^{-1}-\lambda I)=0$. Thus, the reproductive number of Model 1, $R_{0}^{1}$, is as follows:(14)$$R_{0}^{1}=\frac{(1-\bar{\epsilon })p\beta T_{0}}{{c\delta }_{I_{2}}}$$

By substituting the parameters from Table 3, the reproductive number of Model 1, $R_{0}^{1}$, is 335.43.

### 5.2. The Basic Reproductive Number of Model 2

Consider the viral dynamic model with andrographolide and immune response (Model 2) from Equation (5). The kinetic model is defined by using the following equations:(15)$$\begin{array}{c} \frac{dI_{1}}{dt}=\frac{\beta \;TV}{\sigma F+1}-kI_{1},\\ \frac{dI_{2}}{dt}=kI_{1}-\delta_{I_{2}}I_{2}\\ \frac{dV}{dt}=(1-\bar{\epsilon })pI_{2}-cV \end{array}$$

The next generation matrix is given by the following:(16)$${FV}^{-1}=\left(\begin{array}{ccc} 0 & 0 & \frac{\beta T_{0}}{c(\sigma F+1)}\\ 0 & 0 & 0\\ \frac{(1-\bar{\epsilon })p}{\delta_{I_{2}}} & \frac{(1-\bar{\epsilon })p}{\delta_{I_{2}}} & 0 \end{array}\right).$$

Consider $det({FV}^{-1}-\lambda I)=0$. Thus, the reproductive number of Model 2, $R_{0}^{2}$, is as follows:(17)$$R_{0}^{2}=\frac{(1-\bar{\epsilon })p\beta T_{0}}{{c\delta }_{I_{2}}(\sigma F+1)}$$

By substituting the parameters from Table 4, the reproductive number of Model 2, $R_{0}^{2}$, is 335.43.

The calculated $R_{0}$ ≈ 335 refers specifically to the within-host basic reproduction number, representing the average number of newly infected cells generated by one infected cell in the early phase of infection, in the absence of immune control or drug intervention. While this value may appear high compared to epidemiological $R_{0}\;$ estimates, it aligns with reported viral load trajectories from the patient data obtained from a hospital in Thailand [4], where the viral copies increased rapidly within the first three days. Nevertheless, this high value underscores the aggressive replication nature of SARS-CoV-2 and the sensitivity of $R_{0}\;$ to assumptions on viral clearance (c) and infection rates (β). We provide sensitivity analyses to further explore this dependency.

The stability analysis of the proposed viral dynamic models demonstrates robust and predictable behavior under specified parameter conditions. The basic reproductive number ($R_{0}$) is a fundamental threshold parameter that characterizes the average number of new infections generated by a single infected cell or virion in a fully susceptible environment within the host.

Mathematically, $R_{0}<1$ indicates that the virus is eventually cleared, as each infected cell or virion produces less than one new infection on average. Conversely, $R_{0}>1$ signifies that the infection will expand within the host, potentially leading to sustained viral replication and more severe pathology. When $R_{0}=1$, the infection reaches a steady state without self-limiting clearance or exponential growth.

Therefore, we focus on identifying parameter values that ensure $R_{0}<1$. For this model, we have fixed the parameters $T_{0},\;\sigma ,\;k,\;\delta_{I_{2}},\;p,\;\omega$ and g, resulting in the following outcome:(18)$$\frac{(1-\bar{\epsilon })p\beta T_{0}}{{c\delta }_{I_{2}}}<1,$$

Then,(19)$$\frac{(1-\bar{\epsilon })pT_{0}}{\delta_{I_{2}}}<\frac{c}{\beta },$$

By substituting $T_{0}=1.33\times 10^{5},\;\delta_{I_{2}}=0.6,$ and p=27, we obtain $\frac{pT_{0}}{\delta_{I_{2}}}=5.985\times 10^{6}$. Since $0\leq (1-\bar{\epsilon })\leq 1,$ this results in the following:(20)$$0\leq \frac{(1-\bar{\epsilon })pT_{0}}{\delta_{I_{2}}}\leq 5.985\times 10^{6}.$$

Therefore, to satisfy $\;R_{0}<1$, the following must hold:(21)$$\frac{c}{\beta }>5.985\times 10^{6}.$$

In this study, the threshold condition $R_{0}<1$ serves as a guiding criterion for determining the efficacy of therapeutic intervention. Through analytical derivation, we identified a key inequality condition that determines whether viral propagation can be suppressed. Specifically, we show that, if the viral clearance rate c exceeds $5.985\times 10^{6}$ times the infection rate constant β—which governs the conversion of uninfected target cells into infected cells—the infection is unlikely to persist. This result highlights a crucial therapeutic target; enhancing viral clearance mechanisms (e.g., via immune activation or antiviral agents) is more effective for reducing $R_{0}$ than solely suppressing viral entry.

Importantly, this finding contributes a novel quantitative threshold to the field of within-host viral modeling, offering actionable insights for therapeutic design. Unlike previous studies that merely report $R_{0}$ values, this work establishes a specific ratio between mechanistic parameters (clearance vs. infection rate) that can inform dose–response optimization. This ratio can serve as a benchmark for drug development and immune system support strategies aiming to push the system dynamics below the epidemic threshold.

From a public health perspective, the implications extend beyond the individual level. Although $R_{0}$ is traditionally used in population-level epidemiology, its within-host analog can inform treatment timing, dosage strategies, and patient stratification. For instance, patients whose physiological conditions or pharmacokinetic profiles can support a higher viral clearance rate relative to infection rate are more likely to benefit from andrographolide-based therapy. Consequently, the model provides a foundation for personalized medicine approaches where treatment regimens are guided by patient- or population-specific parameter estimates.

In summary, the analysis of $R_{0}\;$ not only reaffirms its role as a stability indicator within the host but also advances the field by introducing a therapeutically actionable threshold that bridges theoretical modeling and clinical application. To our knowledge, this ratio-based interpretation and its direct linkage to drug efficacy thresholds have not been explicitly quantified in previous within-host SARS-CoV-2 modeling studies, representing a significant addition to the literature.

## 6. Conclusions and Discussion

In this study, we employed a mathematical modeling approach to explore the within-host dynamics of SARS-CoV-2 infection under andrographolide treatment. Unlike conventional models that primarily focus on descriptive viral load data, our model integrates immunological feedback and drug intervention variables, offering a systems-level perspective on infection progression and therapeutic modulation. Given that the model is informed by a small dataset derived from only four patients, the findings should be interpreted with caution. This is particularly valuable in contexts where clinical data are limited, such as during early outbreaks or in populations with restricted access to diagnostics. The model serves as an exploratory tool to generate hypotheses and guide future studies involving larger cohorts. Future work should validate the model in larger cohorts and through prospective clinical trials to enhance its applicability and clinical relevance.

Numerical simulations (as illustrated in Figure 5) provide insight into the temporal evolution of viral load and immune responses under varying doses of andrographolide. These results facilitate the identification of potentially optimal treatment windows and inform individualized dosing strategies. Such personalized approaches are increasingly relevant in the era of precision medicine.

Two viral dynamic models were developed to investigate the therapeutic effects of andrographolide. Model 1 exhibited a direct dose–response relationship, indicating that higher andrographolide dosages correlate with a proportional reduction in viral load. In contrast, Model 2 revealed a more complex dynamic, indicating that the efficacy of andrographolide might be influenced by additional variables—possibly pharmacokinetic variability, host immune heterogeneity, or viral escape mechanisms. This divergence between the two models underscores the biological and pharmacological complexity of SARS-CoV-2 infection and suggests the presence of nonlinear interactions that merit further investigation.

PK/PD modeling demonstrated that plasma drug exposure, as measured by both plasma andrographolide concentration and its half-maximal inhibitory concentration (${IC}_{50}$), increases significantly with higher andrographolide dosages [11]. These simulated PK profiles were incorporated into a within-host viral dynamics model to estimate the drug’s antiviral efficacy. Consistent with this, Songvut et al. reported clinical PK data showing that increasing the dose from 180 mg/day to 360 mg/day nearly doubled the Cmax of andrographolide [17].

This pharmacokinetic behavior has important implications for treatment optimization, as it suggests diminishing returns at higher doses and the need for careful balancing of efficacy and safety. Safety data indicate that high-dose andrographolide (180–360 mg/day) is generally well tolerated with mild, reversible adverse effects; however, elevated liver enzymes after 7 days support limiting treatment duration to under 7 days, in line with Thai Herbal Pharmacopoeia guidelines and clinical practice during COVID-19 [17]. Collectively, these findings highlight the critical importance of integrating PK/PD data with dynamic infection models and safety evidence to inform rational dose selection and improve the translational potential of andrographolide in the treatment of SARS-CoV-2 and other respiratory viral infections.

Both models demonstrated internal stability, with computed within-host basic reproductive numbers ($R_{0})$ consistently above zero, affirming the virus’s potential for sustained propagation in the absence of intervention. Here, $R_{0}\;$ refers to the number of newly infected target cells generated by a single infected cell within the host. Importantly, analytical results from the model suggest that if the ratio c/β (where c is the viral clearance rate and β is the infection rate of target cells) exceeds a threshold value of $5.985\times 10^{6}$, then $R_{0}<1$ can be achieved. This threshold illustrates a critical balance—if viral clearance (c) greatly exceeds the infection rate (β), the infection can be controlled. This insight introduces a quantitative benchmark for evaluating antiviral efficacy based on the mechanistic balance between host clearance capacity and viral infectivity. Antiviral treatments may enhance c by promoting viral elimination and reduce β by inhibiting viral entry or replication. To our knowledge, such a parameter-driven threshold has not been previously articulated in the context of andrographolide or other phytocompounds targeting SARS-CoV-2, representing a valuable addition to the field of antiviral modeling. This insight is particularly novel, as it introduces a quantitative benchmark for evaluating antiviral efficacy based on a mechanistic balance between host clearance capacity and viral infectivity. To our knowledge, such a parameter-driven threshold has not been previously articulated in the context of andrographolide or other phytocompounds targeting SARS-CoV-2, representing a valuable addition to the field of antiviral modeling.

Furthermore, our findings underscore the therapeutic potential of andrographolide in mitigating COVID-19 progression. Nevertheless, prior studies have documented significant inter-individual variability in response to treatment, likely influenced by factors such as age, comorbid conditions, genetic background, and immune competence. This variability reinforces the necessity of integrating patient-specific parameters into future models to guide personalized treatment protocols and ensure equitable clinical outcomes across diverse populations.

Given the current gaps in knowledge surrounding andrographolide’s full pharmacological profile, further research into its molecular mechanisms, bioavailability enhancements (e.g., via nanoparticle delivery or structural analogues), and immunological interactions is warranted. Such investigations will not only strengthen its application for COVID-19 but also position andrographolide as a potential broad-spectrum antiviral agent.

In summary, this study presents a mechanistic, hypothesis-generating modeling framework, supporting the potential of andrographolide as a therapeutic candidate for SARS-CoV-2 and related respiratory infections. By integrating within-host viral dynamics with PK/PD and safety data, our model lays a foundation for rational dose optimization and individualized treatment strategies. While the findings are promising, experimental validation through in vitro and in vivo studies, as well as clinical trials, remains essential to confirming translational applicability. Ultimately, this work advances the development of phytocompound-based antivirals within the systems pharmacology paradigm, contributing to targeted and effective antiviral interventions.

## Figures and Tables

**Figure 1 viruses-17-00891-f001:**
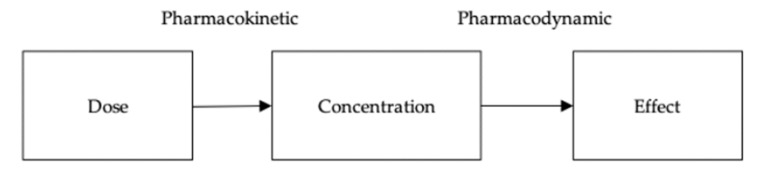
Relationship between drug’s pharmacokinetics and pharmacodynamics.

**Figure 2 viruses-17-00891-f002:**
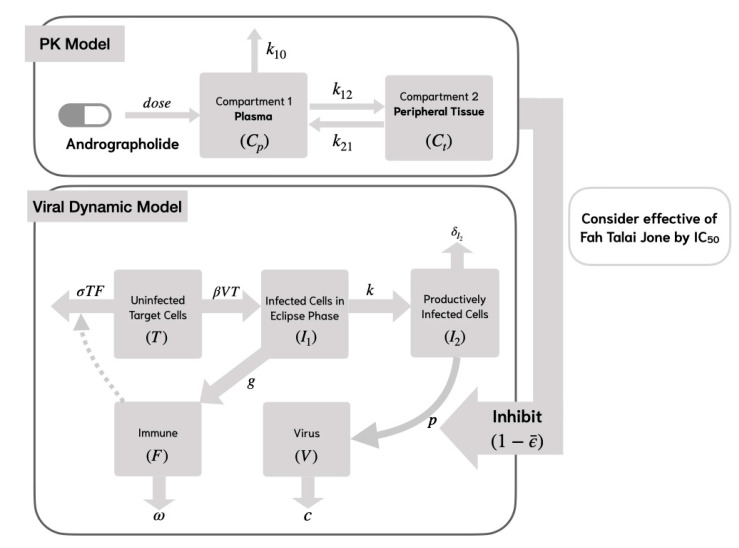
Viral dynamic model of SAR-CoV-2 analyses with andrographolide and nonspecific innate immune response (Model 1).

**Figure 3 viruses-17-00891-f003:**
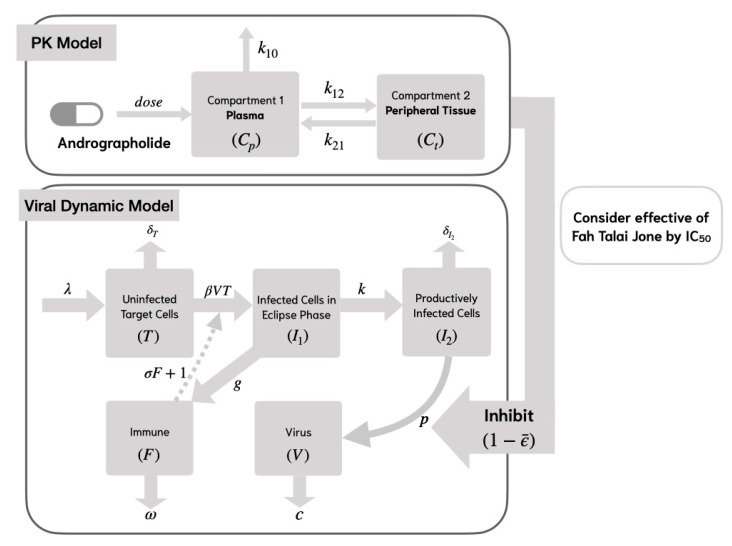
Viral dynamic model of the SAR-CoV-2 analyses with andrographolide and specific lytic and non-lytic immune responses (Model 2).

**Figure 4 viruses-17-00891-f004:**
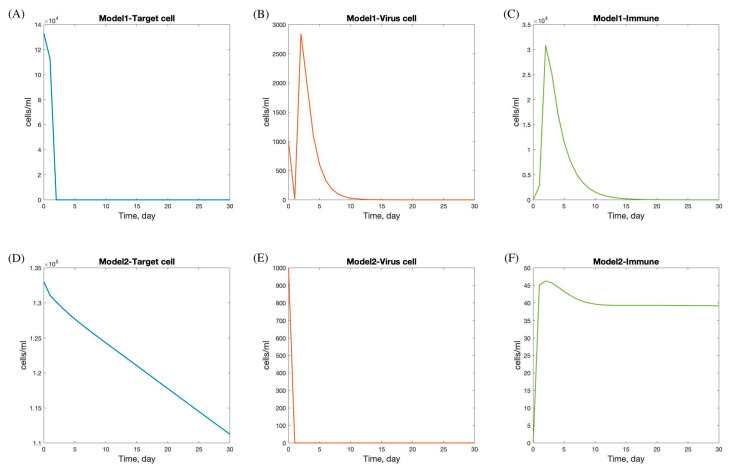
Numerical solution of SAR-CoV-2 dynamic model with andrographolide and immune response. Panels (**A**–**C**) show the dynamics under treatment with andrographolide and immune response (Model 1), including target cells (**A**), virus load (**B**), and immune response (**C**). Panels (**D**–**F**) show the dynamics under treatment with andrographolide and immune response (Model 2), including target cells (**D**), virus load (**E**), and immune response (**F**).

**Figure 5 viruses-17-00891-f005:**
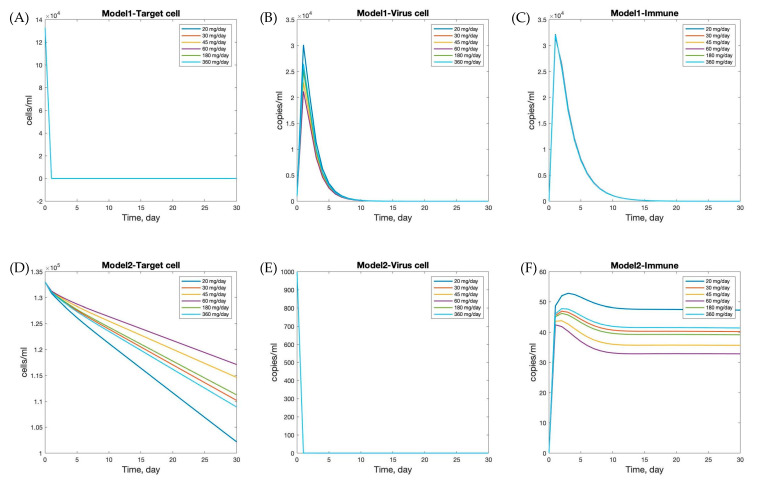
Effects of different andrographolide dosages on within-host dynamics of SARS-CoV-2 infection. Panels (**A**–**C**) show the results of Model 1, and panels (**D**–**F**) show Model 2. Each curve corresponds to a different daily dosage: 20, 30, 45, 60, 180, and 360 mg/day. The plots illustrate the time evolution of target cells (**A**,**D**), virus load (**B**,**E**), and immune cells (**C**,**F**).

**Table 1 viruses-17-00891-t001:** Description of variables and parameters of pharmacokinetic model.

Notations	Description	Unit	Value
$C_{p}(t)$	Plasma concentration of andrographolide at time-*t*	ng/mL	initial value: 0
$C_{t}(t)$	Peripheral tissue concentration of andrographolide at time-*t*	ng/mL	initial value: 0
dose	Dose of administration	ng/mL	estimated
$k_{12}$	The rate at which andrographolide diffuse to peripheral tissue	day^−1^	calculated
$k_{21}$	The rate at which andrographolide diffuse to plasma	day^−1^	calculated
$k_{10}$	The rate at which andrographolide elimination	day^−1^	calculated

**Table 2 viruses-17-00891-t002:** Variables and parameters of viral dynamic model.

Notations	Description	Unit	Value
T(t)	The quantity of target cell at time-*t*	cells/mL	initial value: $1.33\times 10^{5}$ [5]
$I_{1}(t)$	The quantity of infected cell in eclipse phase at time-*t*	cells/mL	initial value: 0
$I_{2}(t)$	The quantity of productively infected cell at time-*t*	cells/mL	initial value: 0
V(t)	The quantity of virion cell at time-*t*	copies/mL	assumed
F(t)	The quantity of immune at time-*t*	cells/mL	initial value: 0
λ	Growth rate of target cells	day^−1^	estimated [10]
$\delta_{T}$	Death rate of target cells	day^−1^	estimated [10]
β	The rate at which circulating virion convert target cells to eclipse phase infected cells	mL/cells/day	estimated
σ	The infected rate which inhibited by non-lytic immune	mL/cells/day	assumed [10]
k	The rate at which infected cells in the eclipse phase convert to productively infected cells	day^−1^	assumed [6]
$\delta_{I_{2}}$	Death rate of productively infected cells	day^−1^	assumed [6]
p	The rate at which productively infected cells produce new virion	day^−1^	assumed [5]
c	Clearance rate of virus	day^−1^	estimated
g	Growth rate of immune	day^−1^	assumed [10]
ω	Death rate of immune	day^−1^	assumed [10]
$\bar{\epsilon }$	The mean effective of andrographolide for 5 days	-	estimated [11]

**Table 3 viruses-17-00891-t003:** Parameter estimation of viral dynamic model (Model 1) with data of patients who were treated with APE.

Patient	β	σ	k	$\delta_{I_{2}}$	p	c	ω	g	$\bar{\epsilon }$
1	$4.20\times 10^{-4}$	$1.97\times 10^{-6}$	3	0.6	27	700	0.4	1	0.6713
2	$8.71\times 10^{-3}$	$1.97\times 10^{-6}$	3	0.6	27	226	0.4	1	0.6713
3	$8.00\times 10^{-2}$	$1.97\times 10^{-6}$	3	0.6	27	50	0.4	1	0.6713
4	$9.00\times 10^{-2}$	$1.97\times 10^{-6}$	3	0.6	27	75	0.4	1	0.6713
Average	$4.48\times 10^{-2}$	$1.97\times 10^{-6}$	3	0.6	27	262.75	0.4	1	0.6713

**Table 4 viruses-17-00891-t004:** Parameter estimation of viral dynamic model (Model 2) with data of patients who were treated with APE.

Patient	λ	$\delta_{T}$	β	σ	k	$\delta_{I_{2}}$	p	c	ω	g	$\bar{\epsilon }$
1	10	0.1	$4.20\times 10^{-4}$	$1.97\times 10^{-6}$	3	0.6	27	700	0.4	1	0.6713
2	10	0.1	$8.71\times 10^{-3}$	$1.97\times 10^{-6}$	3	0.6	27	226	0.4	1	0.6713
3	10	0.1	$8.00\times 10^{-2}$	$1.97\times 10^{-6}$	3	0.6	27	50	0.4	1	0.6713
4	10	0.1	$9.00\times 10^{-2}$	$1.97\times 10^{-6}$	3	0.6	27	75	0.4	1	0.6713
Average	10	0.1	$4.48\times 10^{-2}$	$1.97\times 10^{-6}$	3	0.6	27	262.75	0.4	1	0.6713

**Table 5 viruses-17-00891-t005:** Average effectiveness of andrographolide in treating COVID-19 [11].

Dosage (mg/mL)	The Average Effectiveness of Andrographolide, $\bar{\epsilon }$
30	0.6630
45	0.6990
60	0.7211
180	0.6713
360	0.6534

## Data Availability

The original contributions presented in the study are included in the article, further inquiries can be directed to the corresponding authors.

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
