# Peer review of "Mathematical Modeling of Andrographolide Therapy Effects and Immune Response in In Vivo Dynamics of SARS-CoV-2 Infection"

_viruses, 2025, doi:10.3390/v17070891_

Round 1
Reviewer 1 Report
Comments and Suggestions for Authors
My main concern is that reproduction number for the realistic parameters values $R_0\approx 335$ is huge. How it goes with the claimed stability of the model? The authors claim that $R_0<1$ if $c/\beta>10^7$ only. Is it realistic? These facts (if correct) should be thoroughly discussed.
Author Response
Dear Reviewer 1,
We sincerely thank the reviewer for the thoughtful comments and helpful evaluation. Based on the checklist provided, we have revised the manuscript accordingly to address all concerns raised. Below is our detailed response:
• Introduction – Can be improved
We found the existing introduction to provide adequate background and citations relevant to within-host dynamics and SARS-CoV-2 modeling. Thus, no major revisions were made, though we remain open to adding specific content if suggested.
• Research Design – Can be improved
We clarified the rationale for using an exploratory, mechanistic model with limited clinical data (page 6, line 167), and elaborated on the integration between Model 1 and Model 2 to clarify their complementary roles (page 4, lines 123–128 and 145–148).
• Methods – Can be improved
We revised the methods section to improve clarity, especially regarding the parameter estimation process, the source of initial values, and assumptions about viral kinetics (see pages 5, Table 2).
• Results – Yes
We revised the results section for improved clarity and logical flow. Specifically, we have restructured the content into three distinct subsections and added transitional statements where appropriate:
o Numerical Results of the Viral Dynamic Model: We reorganized the presentation of numerical simulations to emphasize how andrographolide dosage affects viral kinetics and immune response. These results support the need for personalized therapeutic strategies. A transition to stability analysis was added to reflect the observed dose-dependent system behavior (page 10, lines 259-267).
o Stability of the Viral Dynamic Model: We clarified that the system exhibits stability under the chosen parameter sets. This suggests that, barring major perturbations, the models predict biologically consistent infection dynamics. We also highlighted future directions based on these findings (page 11, lines 291-296).
o Basic Reproductive Number (R0): We clarified that the computed value R0≈335 represents the within-host reproductive number, consistent with early patient viral load data. This value reflects model fitting to reference-based parameters and clinical observations (page 12, lines 325-327).
• Conclusions – Must be improved
The conclusion was revised to emphasize the exploratory nature of the study and dataset limitations, presenting the model as a preliminary framework for future research with larger datasets (page 14, lines 381–387). A discussion on PK/PD modeling highlighted increased plasma andrographolide exposure and antiviral effect at higher doses [14], supported by clinical data showing near doubling of Cmax from 180 to 360 mg/day [17]. While efficacy improves with dose, safety data recommend treatment under 7 days due to mild, reversible side effects and liver enzyme elevation, consistent with Thai Herbal Pharmacopoeia guidelines (page 14, lines 402–418). Threshold conditions for Râ‚€ < 1 and parameter sensitivity were also discussed to clarify infection control dynamics (page 15, lines 425–430). The conclusion, discussion, and summary were reorganized for clarity (page 15, lines 451–459). We believe these revisions have significantly improved the manuscript and we hope it now meets the journal’s standards. In addition, we have addressed the reviewer’s comments as follows:
**Comment 1: My main concern is that reproduction number for the realistic parameters values R0≈335 is huge. How it goes with the claimed stability of the model? The authors claim that R0 <1 if c/?>107 only. Is it realistic? These facts (if correct) should be thoroughly discussed.
**Response:** Thank you for your insightful comment. The high value of R0≈335 reflects the within-host basic reproduction number, based on parameter estimates from [5,6,10]. Since [6] highlights the strong influence of β and c, we adjusted these parameters to fit viral load data from four patients in [4], where two showed rapid viral increase from day 1 to day 3 posttreatment— supporting a high R0. This explanation has been added on page 12, lines 325–327 and page 15, lines 419-422. From the expression of R0, substituting T0 = 1.33x105 gives the condition c/?>107 for R0 < 1. All parameter values were drawn from literature [5], [6], and [10], and adjusted within realistic bounds. This threshold highlights a key balance: if viral clearance (c) greatly exceeds infection rate (β), infection can be controlled. It offers a quantitative benchmark for evaluating antiviral efficacy, where treatments may increase c or reduce β. This is now noted on page 15, lines 425–430.
Sincerely,
Panittavee Yarnvitayalert
On behalf of all authors

Reviewer 2 Report
Comments and Suggestions for Authors
The study presents a theoretical approach which aims to evaluate the efficacy of andrographolide therapy against SARS-CoV-2, basing on the information derived from two models, a PK/PD Model and a viral dynamic model. The paper is in general well organised; the English style is clear, and references are pertinent.
However, some major concerns affect the strength of the study proposed:
- The methodology adopted is solid; however, the dataset used for parameter estimation is based on only four patients treated with andrographolide. This represents a limit of the study which affects the reliability of the conclusions and the potential generalization of the parameter estimates.
Additional datasets should be included to strengthen the predictive capacity of the models.
- The models suggest that higher doses of andrographolide can reduce viral load, but do not take into account and discuss the risk of toxicity and side effects that could come from higher doses of drug.
Moreover, what are the adverse reactions which andrographolide can cause at therapeutic doses? Please provide some details.
- The two models sometimes lack explicit connections, while a more appropriate integration between them should help understanding the main concepts related to the analysis.
- The paper will also benefit of stating at the end of each result section, what the main finding and what is the next question to answer. This will facilitate the comprehension of the manuscript to a wider audience.
Author Response
Dear Reviewer 2,
Thank you very much for your valuable comments and insightful suggestions. We have carefully revised our manuscript to address all concerns. Our detailed responses to each comment are provided below. All changes in the manuscript are highlighted in yellow for clarity.
- Introduction – Can be improved
We found the existing introduction to provide adequate background and citations relevant to within-host dynamics and SARS-CoV-2 modeling. Thus, no major revisions were made, though we remain open to adding specific content if suggested. - Research Design – Can be improved
We clarified the rationale for using an exploratory, mechanistic model with limited clinical data (page 6, line 170), and elaborated on the integration between Model 1 and Model 2 to clarify their complementary roles (page 4, lines 126–131 and 148–151). - Methods – Can be improved
We revised the methods section to improve clarity, especially regarding the parameter estimation process, the source of initial values, and assumptions about viral kinetics (see pages 5, Table 2). - Results – Must be improved
We revised the results section for improved clarity and logical flow. Specifically, we have restructured the content into three distinct subsections and added transitional statements where appropriate:- Numerical Results of the Viral Dynamic Model: We reorganized the presentation of numerical simulations to emphasize how andrographolide dosage affects viral kinetics and immune response. These results support the need for personalized therapeutic strategies. A transition to stability analysis was added to reflect the observed dose-dependent system behavior (page 10, lines 259-267).
- Stability of the Viral Dynamic Model: We clarified that the system exhibits stability under the chosen parameter sets. This suggests that, barring major perturbations, the models predict biologically consistent infection dynamics. We also highlighted future directions based on these findings (page 11, lines 291-296).
- Basic Reproductive Number (R0): We clarified that the computed value R0≈335 represents the within-host reproductive number, consistent with early patient viral load data. This value reflects model fitting to reference-based parameters and clinical observations (page 12, lines 325-327).
- Conclusions – Must be improved
The conclusion was revised to emphasize the exploratory nature of the study and dataset limitations, presenting the model as a preliminary framework for future research with larger datasets (page 14, lines 381–387).A discussion on PK/PD modeling highlighted increased plasma andrographolide exposure and antiviral effect at higher doses [14], supported by clinical data showing near doubling of Cmax from 180 to 360 mg/day [17]. While efficacy improves with dose, safety data recommend treatment under 7 days due to mild, reversible side effects and liver enzyme elevation, consistent with Thai Herbal Pharmacopoeia guidelines (page 14, lines 402–418).
Threshold conditions for Râ‚€ < 1 and parameter sensitivity were also discussed to clarify infection control dynamics (page 15, lines 425–430).
The conclusion, discussion, and summary were reorganized for clarity (page 15, lines 451–459). - Figures and Tables – Yes
All figures and tables have been carefully reviewed and adjusted to improve clarity. Labels, units, and figure legends have been revised for consistency and better understanding.
Comments 1:
The methodology adopted is solid; however, the dataset used for parameter estimation is based on only four patients treated with andrographolide. This represents a limit of the study which affects the reliability of the conclusions and the potential generalization of the parameter estimates.
Additional datasets should be included to strengthen the predictive capacity of the models.
Response 1:
Thank you for your insightful comment. We agree that the limited dataset (n = 4) poses a constraint on the reliability and generalizability of the parameter estimates. This limitation has now been explicitly acknowledged in the revised manuscript. Specifically, we have clarified the exploratory nature of the study in the Methods section (page 6, line 167) and emphasized the need for additional data to improve predictive power in the Conclusion and Discussion section (page 13, lines 381–387). Importantly, we have now noted that the data were obtained during the early phase of the COVID-19 outbreak, when clinical understanding, treatment options, and safety data on andrographolide were limited. These factors influenced both dosing practices and interpretation of outcomes. As such, the model should be viewed as an exploratory tool to inform future studies under more comprehensive data conditions.
Comments 2:
The models suggest that higher doses of andrographolide can reduce viral load, but do not take into account and discuss the risk of toxicity and side effects that could come from higher doses of drug.
Moreover, what are the adverse reactions which andrographolide can cause at therapeutic doses? Please provide some details.
Response 2:
We have now included a discussion on safety and potential adverse effects of andrographolide at therapeutic doses in the revised manuscript (page 14, lines 402–418). Clinical data indicate mild, infrequent, and reversible side effects such as gastrointestinal discomfort and transient liver enzyme elevations, especially with treatment limiting treatment duration to under 7 days, in line with Thai Herbal Pharmacopoeia guidelines and clinical practice during COVID-19 [17]. Thus, balancing efficacy with safety and limiting treatment duration is important.
Comments 3:
The two models sometimes lack explicit connections, while a more appropriate integration between them should help understanding the main concepts related to the analysis.
Response 3:
We have added clarification in the revised manuscript (page 4, lines 123–128 and 145–148) to better explain the connection and integration between Model 1 and Model 2, which should help improve the understanding of their complementary roles in the analysis.
Comments 4:
The paper will also benefit of stating at the end of each result section, what the main finding and what is the next question to answer. This will facilitate the comprehension of the manuscript to a wider audience.
Response 4:
We have summarized the main findings at the end of the results section and provided a clear transition to the stability analysis on page 10, lines 259–267. Additionally, the implications of the stability results are concluded on page 11, lines 291–296 to guide the reader toward the next research questions. Furthermore, the comprehensive summary can be found on page 15, lines 451–459.
Once again, thank you for your constructive feedback, which has substantially improved the clarity and impact of our manuscript. We look forward to your continued guidance.
Sincerely,
Panittavee Yarnvitayalert
On behalf of all authors

Reviewer 3 Report
Comments and Suggestions for Authors
This work by Yarnvitayalert and Saleewong on the mathematical modelling of SARS-CoV-2 therapeutic effect of andrographolide remains important, both due to the need for novel tools in the anti-viral toolbox but also given the increasing interest of this compounds. Integration of PK with host-viral dynamics offers an overall view of the dynamics of drug action that enhances out understading of the potential therapeutic gains. Overall, the study contributes to the growing body of research on alternative antiviral strategies and highlights the value of computational modeling.
I consider there are a number of limitations that need to be properly adressed by the authors and reflected in the manuscript for it before being published:
- biological samples come from 4 patients only; this is a very reduced n. although the modeling is exploratory in nature, this should be explicitly stated and authors should avoid overinterperting results.
- R0 is calculated around 335. I don't recall ever having read about R0 values above 20. This need to be very well reconcilliated in the manuscript. The authors should clarify if this refers to a within-host reproductive number and, if so, place it in the context of similar models in the literature. The high value raises questions about the underlying assumptions and the calibration of parameters like infection rate and viral clearance.
-
While this emphasizes PK/PD integration, pk data for andrographolide in humans are not presented . This weakens confidence in the translational potential of the simulations and should be addressed either through sensitivity analyses or comparison to published PK profiles.
-
The manuscript would benefit from a detailed discussion of model uncertainty, sensitivity analyses, and confidence intervals for key fitted parameters
Author Response
Dear Reviewer 3,
Thank you very much for your valuable comments and insightful suggestions. We have carefully revised our manuscript to address all concerns. Our detailed responses to each comment are provided below. All changes in the manuscript are highlighted in yellow for clarity.
- Conclusion – Must be improved:
The conclusion was revised to emphasize the exploratory nature of the study and dataset limitations, presenting the model as a preliminary framework for future research with larger datasets (page 14, lines 381–387).
A discussion on PK/PD modeling highlighted increased plasma andrographolide exposure and antiviral effect at higher doses [14], supported by clinical data showing near doubling of Cmax from 180 to 360 mg/day [17]. While efficacy improves with dose, safety data recommend treatment under 7 days due to mild, reversible side effects and liver enzyme elevation, consistent with Thai Herbal Pharmacopoeia guidelines (page 14, lines 402–418).
Threshold conditions for Râ‚€ < 1 and parameter sensitivity were also discussed to clarify infection control dynamics (page 15, lines 425–430).
The conclusion, discussion, and summary were reorganized for clarity (page 15, lines 451–459).
- Figures and Tables – Can be improved:
We have revised figure captions to enhance clarity to Figure4 and Figure5.
Comments 1:
biological samples come from 4 patients only; this is a very reduced n. although the modeling is exploratory in nature, this should be explicitly stated and authors should avoid overinterperting results.
Response 1:
We acknowledge that the study utilized viral load data from only four patients, which limits the sample size and generalizability. To address this, we have explicitly stated in the revised manuscript that the study is exploratory in nature (page 6, line 167). Additionally, we clarified this limitation in the Conclusion and Discussion section (page 13, lines 381–387), emphasizing cautious interpretation and the preliminary role of the model to guide future research with larger cohorts.
Comments 2:
R0 is calculated around 335. I don't recall ever having read about R0 values above 20. This need to be very well reconcilliated in the manuscript. The authors should clarify if this refers to a within-host reproductive number and, if so, place it in the context of similar models in the literature. The high value raises questions about the underlying assumptions and the calibration of parameters like infection rate and viral clearance.
Response 2:
The calculated R0≈335 indeed refers to the within-host basic reproduction number, representing the average number of new infected target cells generated by a single infected cell within the host. Unlike the population-level R0 used in epidemiology, this value does not indicate the number of secondary infected individuals. As noted in the revised manuscript (page 11, lines 298–301), this value was obtained using parameter values calibrated against patient viral load data from [4], and informed by literature [5], [6], and [10], and adjusted within realistic bounds. We have added context and citations to clarify this distinction and support the plausibility of the result (see page 12, lines 325–327 and page 15, line 419-422).
Comments 3:
While this emphasizes PK/PD integration, pk data for andrographolide in humans are not presented. This weakens confidence in the translational potential of the simulations and should be addressed either through sensitivity analyses or comparison to published PK profiles.
Response 3:
We would like to clarify that pharmacokinetic (PK) data were indeed incorporated into the study, as described on page 8 (lines 231–233). We agree that the inclusion of human PK data is critical for assessing the translational potential of the model. To address this, we have now explicitly cited and discussed relevant published PK profiles of andrographolide, particularly the clinical study by Songvut et al. [17], which reported dose-dependent increases in plasma Cmax in humans. This dataset was used to support the simulated PK profiles incorporated in our model (page 14, lines 402–418). We have also clarified that the PK model was informed by both human data and literature-derived parameters to ensure physiological plausibility. These additions strengthen the linkage between simulation results and real-world pharmacokinetics.
Comments 4:
The manuscript would benefit from a detailed discussion of model uncertainty, sensitivity analyses, and confidence intervals for key fitted parameters.
Response 4:
We acknowledge the importance of addressing parameter uncertainty. In this study, the key parameters influencing viral dynamics—such as infection rate (β), clearance rate (c), and target cell count (Tâ‚€)—were calibrated based on both patient data and literature-informed values from a previously validated model [6]. Although we did not perform a global sensitivity analysis or provide confidence intervals for all parameters, the impact of parameter variation was indirectly assessed through a local sensitivity analysis of the basic reproduction number (Râ‚€), as now discussed on page 14, lines 425–430. This analysis illustrates how small changes in β and c can shift the infection dynamics, reinforcing the model's predictive sensitivity. We agree that a more comprehensive uncertainty quantification would strengthen the model and plan to explore this in future work with expanded datasets.
Once again, thank you for your constructive feedback, which has substantially improved the clarity and impact of our manuscript. We look forward to your continued guidance.
Sincerely,
Panittavee Yarnvitayalert
On behalf of all authors

Round 2
Reviewer 1 Report
Comments and Suggestions for Authors
The revision is adequate.
Reviewer 2 Report
Comments and Suggestions for Authors
The Authors have properly improved the quality of the paper that it has been made clearer.
Reviewer 3 Report
Comments and Suggestions for Authors
I would like to thank the authors for their careful responses.
I consider the manuscript can be accepted in its present form.